# Can we spot fake public comments generated by ChatGPT(-3.5, -4)?: Japanese stylometric analysis expose emulation created by one-shot learning

**Wataru Zaitsu**[1¤]*, **Mingzhe Jin**[2], **Shunichi Ishihara**[3], **Satoru Tsuge**[4], **Mitsuyuki Inaba**[5]

**1** Faculty of Psychology, Mejiro University, Tokyo, Japan, **2** Institute of Interdisciplinary Research, Kyoto University of Advanced Science, Kyoto, Japan, **3** Speech and Language Laboratory, Australian National University, Canberra, Australia, **4** School of Informatics, Daido University, Aichi, Japan, **5** College of Policy Science, Ritsumeikan University, Kyoto, Japan

¤ Current address: Department of Psychological Counselling, Faculty of Psychology, Mejiro University, Tokyo, Japan
* w.zaitsu@mejiro.ac.jp

**Data Availability Statement:** All relevant data are within the Supporting information files.

**Funding:** This work was partially supported by JSPS KAKENHI (grant number: JP23K11107). The

## Abstract

Public comments are an important opinion for civic when the government establishes rules. However, recent AI can easily generate large quantities of disinformation, including fake public comments. We attempted to distinguish between human public comments and ChatGPT-generated public comments (including ChatGPT emulated that of humans) using Japanese stylometric analysis. Study 1 conducted multidimensional scaling (MDS) to compare 500 texts of five classes: Human public comments, GPT-3.5 and GPT-4 generated public comments only by presenting the titles of human public comments (i.e., zero-shot learning, $GPT_{zero}$), GPT-3.5 and GPT-4 emulated by presenting sentences of human public comments and instructing to emulate that (i.e., one-shot learning, $GPT_{one}$). The MDS results showed that the Japanese stylometric features of the public comments were completely different from those of the $GPT_{zero}$-generated texts. Moreover, $GPT_{one}$-generated public comments were closer to those of humans than those generated by $GPT_{zero}$. In Study 2, the performance levels of the random forest (RF) classifier for distinguishing three classes (human, $GPT_{zero}$, and $GPT_{one}$ texts). RF classifiers showed the best precision for the human public comments of approximately 90%, and the best precision for the fake public comments generated by GPT ($GPT_{zero}$ and $GPT_{one}$) was 99.5% by focusing on integrated next writing style features: phrase patterns, parts-of-speech (POS) bigram and trigram, and function words. Therefore, the current study concluded that we could discriminate between GPT-generated fake public comments and those written by humans at the present time.

## Introduction

Currently, we are facing an unprecedented crisis caused by artificial intelligence (AI). The proliferation of disinformation such as fake news and images may begin to surround us without

funders had no role in the study design, data collection, analysis, and decision to publish, except for the Publication Fee.

**Competing interests:** We have declared that no competing interests exist.

our recognition. ChatGPT [1] has played a major role in sparking the beginning. This large language model (LLM), trained and released by OpenAI on November 30, 2022, naturally generates human-like text. Recent chatbots have a generative pretrained transformer (GPT), which dramatically improves the generative performance. These chatbots are convenient and provide various benefits, but it is easy to imagine many kinds of problems, such as manipulating public opinion, writing fake customer reviews, and submitting fabricated academic papers. It has already become possible for anyone to easily generate a large amount of fake public comments for the purpose of making the government create laws and regulations in line with one's own opinions. To make matters worse, previous studies [2, 3] have verified that almost no people can distinguish between AI-generated and human-written sentences at first glance. Such social problems have already arisen worldwide. Therefore, controlling and understanding generative AI is an urgent issue for humans. The purpose of this study is to try to classify human public comments and ChatGPT-generated fake public comments.

Several researchers have reported the possibility of distinguishing between ChatGPT-generated and human-written texts [4, 5]. Desaire et al. [4] made ChatGPT-3.5 learn human-written academic papers as training data and compared ChatGPT-generated and human-written texts. Zaitsu & Jin [5] also gave instructions against ChatGPT to generate texts by presenting the titles of Japanese scientific academic papers. The results of these studies were distinguishable with nearly 100% accuracy. However, several studies for distinguishing AI-generated and human-written sentences exist. Therefore, it is necessary to conduct research that targets various genres. Brown et al. [6] proposed a learning method without changing the parameters of GPT-3, such as fine-tuning: zero-shot, one-shot, and few-shot learning. One-shot or few-shot learning attempts to obtain an answer by providing prompts with any additional information, whereas zero-shot learning only provides instructions without other information. A question arises here: when we present human-written text as a sample against AI and instruct them to emulate the contents and writing styles of the example, can we distinguish between AI-emulated and human-written texts? In this study, we compared ChatGPT-generated fake public comments with and without emulation (i.e., zero-shot or one-shot learning) to human-written true public comments. This study will make a great contribution to the solutions of problems and risks facing modern society, especially manipulating public opinions using fake public comments generated by ChatGPT.

Public comments (or public consultations) are important civic opinions in establishing rules and orders, such as laws and regulations, and differ from academic papers in two ways: (1) Higher degree of freedom in writing styles because public comments have fewer constraints. (2) Public comments (a few hundred characters) have fewer word counts than academic papers (over thousands of characters). It is expected that the higher the degree of freedom for writing, the easier it is to discriminate the texts of both AI and humans because the features of writing styles are easily expressed. On the other hand, the fewer the word counts, the more difficult the discrimination because the amount of information available for distinguishing decreases.

In study 1, we prepared sample through following methods: (1) "HM" texts (100 samples): human public comments published by Japanese national administrative agencies, (2) "GPT3.5$_{zero}$" or "GPT4$_{zero}$" texts (every 100 samples): ChatGPT (GPT-3.5 and -4)-generated texts with only presenting the title of public comments (zero-shot learning), (3) "GPT3.5$_{one}$" or "GPT4$_{one}$" texts (each 100 sample): We instructed ChatGPT (GPT-3.5 and -4) to emulate the contents and writing styles of human public comments while presenting the entire body (one-shot learning). Each fake public comment generated by both ChatGPT and each public comment text written by a human were paired and had similar content. Next, we compared these texts from the perspective of their Japanese stylometric features. Especially, we analyzed

no meaning stylometric features such as function words or sentence structures, rather than content words such as noun 'cat', verb 'run', and adjective 'beautiful', because the former features are not dependent on topic and genre of texts.

Thus, this study proposes the following hypothesis: Hypothesis 1: As shown in a previous study [5], the Japanese stylometric features of both $GPT_{zero}$ texts ($GPT3.5_{zero}$ and $GPT4_{zero}$) are completely different from those of HM texts, even in public comments. Hypothesis 2: Both $GPT_{one}$ texts ($GPT3.5_{one}$ and $GPT4_{one}$) are closer to the HM texts than the $GPT_{zero}$ texts because of the effect of one-shot learning. Hypothesis 3: We can discriminate GPT-generated both types of fake public comments (both $GPT_{zero}$ and $GPT_{one}$) from human public comments using Japanese stylometric analysis, even if this study supposed hypothesis 2.

## Method

### Sample

As stated previously, we collected 100 Japanese public comments from the e-Gov website (https://www.e-gov.go.jp) published by Japanese national administrative agencies. There is no copyright problem because this website states that published information is not subject to copyright and can be freely used. Public comments covered various topics: telework security guidelines, eel aquaculture, support for the independence of the homeless, personal information protection law, etc. The number of characters in HM texts resulted in a mean of 661.3 (*SD* 132.0) and a median of 627.

Next, we make ChatGPT generated 100 texts (GPT-3.5) and 100 texts (GPT-4) in Japanese (i.e., $GPT3.5_{zero}$ and $GPT4_{zero}$ texts) with the next prompts: "You are 'general citizen.' Write a public comment (criticism, request, and opinion) about 'title of the public comment.'." If the attribute of the person who wrote the public comment was known to us, we change 'general citizen' to a specified attribute such as business person, lawyer, or doctor. The number of characters showed a mean of 604.3 (*SD* 61.3) and a median of 601.5 in $GPT3.5_{zero}$ and a mean of 620.4 (*SD* 61.8) and a median of 621 in $GPT4_{zero}$.

Lastly, as with $GPT3.5_{one}$ and $GPT4_{one}$ texts, we have ChatGPT generated two sets of 100 Japanese texts by having each ChatGPT (-3.5 and -4) emulate while presenting human public comments with the next prompts: "The following statement is a public comment (criticism, request, and opinion) submitted from a general citizen. Write a public comment similar in content and in writing style to this statement." The number of characters of $GPT3.5_{one}$ was a mean of 603.3 (*SD* 71.5) and a median of 594 and that of $GPT4_{one}$ was a mean of 604.6 (*SD* 54.8) and a median of 621.

### Japanese stylometric features

We counted the frequency of occurrence of the next stylometric features and calculated the rate of frequency of occurrence within each text to avoid depending on the length of the count words of the texts.

**Phrase patterns.** Phrase patterns are regarded as effective features for authorship attribution in the Japanese language [7]. To analyze these features, we attached POS tags to each word using morphological analysis and divided the sentences into phrases using syntactic analysis. After the analysis, we focused on the combination of function words and POS of content words within each phrase: "noun + が (postpositional particle)", "noun + noun + へ (postpositional particle) + の (postpositional particle)", "実際 (adverb) + に (postpositional particle)", and "noun + noun + noun + の (postpositional particle)" etc.

**Parts-of-speech (POS) bi- and trigrams.** The concept of *N*-gram is used in the field of quantitative linguistics to determine the frequency of a contiguous sequence of symbols

(characters, words, phrases, etc.) in a sentence. Bigram is in the case of $N = 2$ ("preposition + noun" etc.), and trigram is in the case of $N = 3$ ("preposition + noun + adjective" etc.). Both POS trigrams and bigrams are effective stylometric features for authorship attribution [8].

**Bigrams of postpositional particle words.** The frequency of a contiguous sequence of postpositional particle words such as "を(case particle) + の (case particle)" and "は (binding particle) + が (case particle)" etc. A previous study on Japanese authorship attribution [9] reported effectiveness as a distinguishable feature but lower performance in AI detection tasks [5].

**Positioning of commas.** Positioning of commas is where the author used commas in sentences such as "は (binding particle) +,(comma)," "する (verb) +,(comma)," and "だ (auxiliary verb) +,(comma)." In other words, we focused on the words before the comma.

**Function words.** Preceding study of authorship attribution [10] and AI detection task [5] reported the function words as quite distinguishable features: "だ (auxiliary verb)," "また (conjunction)," and "は (postpositional particle)."

In Study 1, we confirmed which stylometric features were effective; in Study 2, we consolidated the effective features into integrated ones to examine incremental validity in verifying distinguishable performance levels.

In morphological analysis, we used the Japanese POS tagger Mecab [11] and attached POS tags (e.g., postpositional particle: "case particle," "binding particle," and "ending particle"). When syntactic analysis was conducted, we used the Japanese parser CaboCha [12].

## Analysis procedure

The current study essentially adopted the analysis procedure and statistical methods of Zaitsu & Jin [5] to compare the current results with prior results.

**Study 1.** To examine Hypotheses 1 and 2, we used classical multidimensional scaling (MDS). This statistical method can display the similarity between texts as distance; the more similar both texts are, the closer they are in dimensions. In MDS, the definitions of distances exist in various forms, and we used the symmetric Jensen-Shannon divergence distance ($d_{SJSD}$) to compare 500 texts of five classes (HM, GPT3.5$_{zero}$, GPT4$_{zero}$, GPT3.5$_{one}$, and GPT4$_{one}$) in each Japanese stylometric feature because it is effective for authorship attribution [13] and AI detection [5]. The Eq (1) for the distance between x and y is shown below. We conducted MDS using the *cmdscale* function of the **stats** package of the R language.

$$d_{SJSD}(\boldsymbol{x}, \boldsymbol{y})^2 = \frac{1}{2} \Sigma_{i=1}^{n} \left( x_i \log \frac{2x_i}{x_i + y_i} + y_i \log \frac{2y_i}{x_i + y_i} \right) \tag{1}$$

**Study 2.** To verify the performance level for distinguishing among the three classes (GPT$_{zero}$, GPT$_{one}$, and HM), we used random forest (RF) and executed leave-one-out cross-validation (LOOCV). The RF classifier is a classical machine learning method similar to bagging. The reasons that we selected this classifier are follows: (1) The RF classifier is effective for authorship attribution [14] among several other classifiers and AI detection [5] in Japanese. (2) we investigate the effective stylometric features for distinguishing AI-generated texts from human-written ones. LOOCV is a type of cross-validation used to evaluate the generalization performance of a model. In this study, one text was excluded from the 500 texts as the testing set, and the RF classifier was trained using the remaining 499 texts to classify the testing text into one of three classes. These procedures were repeated 500 times using different test sets. We used the *randomForest* function of the **random Forest** package and set the number of decision trees to 1,000 and the other hyperparameters to default.

## Results

### Study 1: Comparison of text distributions of five classes (GPT3.5$_{zero}$, GPT4$_{zero}$, GPT3.5$_{one}$, and GPT4$_{one}$, HM)

Figs 1–6 show the degrees of similarity and difference between the texts belonging to the five different classes separately for the six types of stylometric features. First, except for the positioning of commas in Fig 5, the stylometric features (Figs 1–4 and 6) appear to be HM texts that are completely separated from both GPT$_{zero}$ texts. These results support hypothesis 1. Second, all but Fig 5 indicated that GPT3.5$_{zero}$ and GPT4$_{zero}$ have different distributions. Finally, according to all Figures except Fig 5, the distributions of both GPT3.5$_{one}$ and GPT4$_{one}$ are slightly closer to HM texts and are positioned between the distribution of GPT$_{zero}$ texts and that of HM texts. Moreover, some GPT$_{one}$ texts overlapped with HM texts.

Table 1 shows the means and standard deviations of the distances of the texts between GPT (GPT$_{zero}$ and GPT$_{one}$) and HM, corresponding to Figs 1–6. The distances between GPT$_{one}$ and

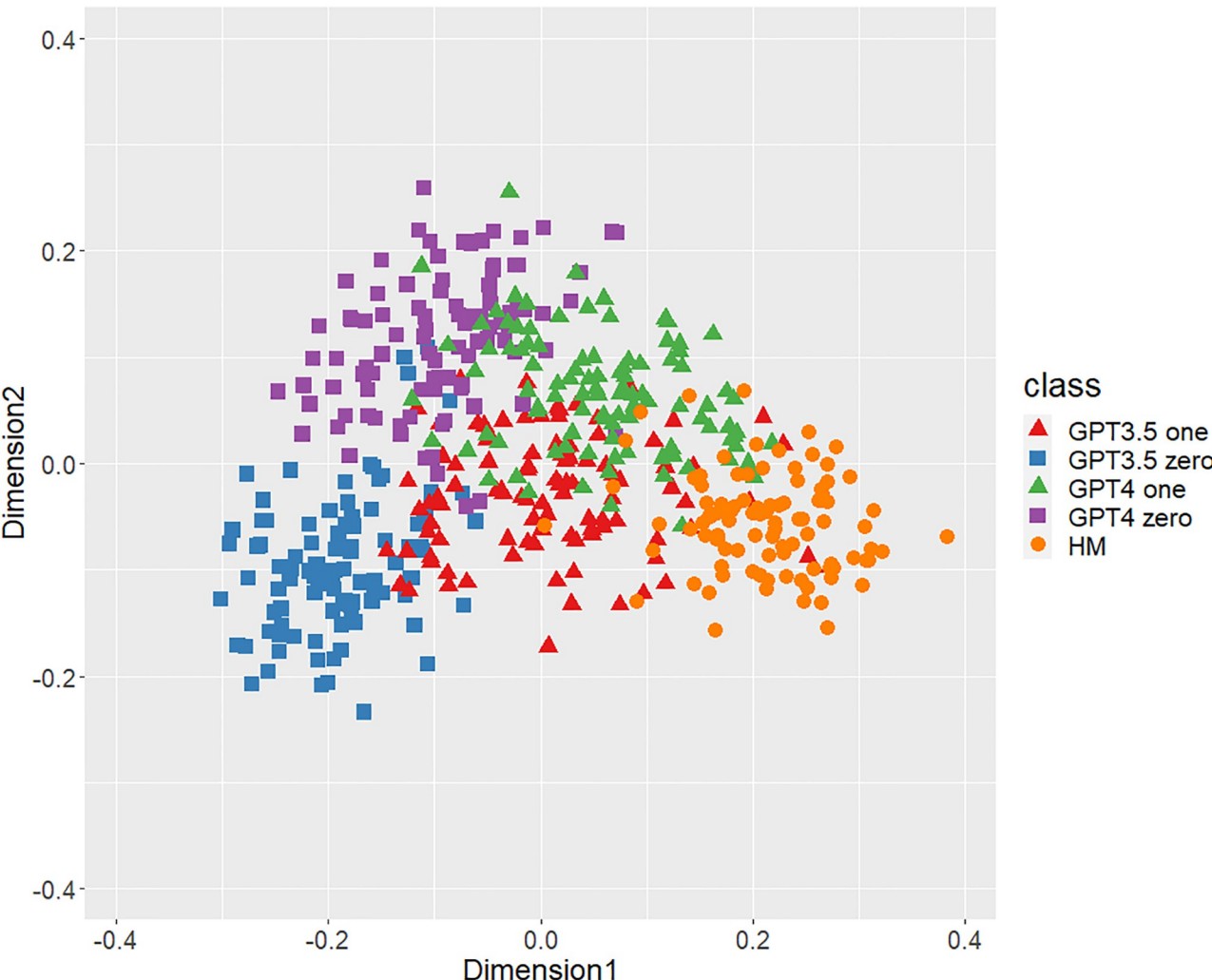

**Fig 1. MDS configuration in five classes (GPT3.5$_{one}$, GPT3.5$_{zero}$, GPT4$_{one}$, GPT4$_{zero}$, and HM), focusing on the phrase patterns.** "GPT3.5$_{one}$" and "GPT4$_{one}$" mean texts generated by GPT-3.5 and GPT-4 with one-shot learning. "GPT3.5$_{zero}$" and "GPT4$_{zero}$" indicate texts generated by GPT-3.5 and GPT-4 with zero-shot learning. "HM" means human-written public comment.

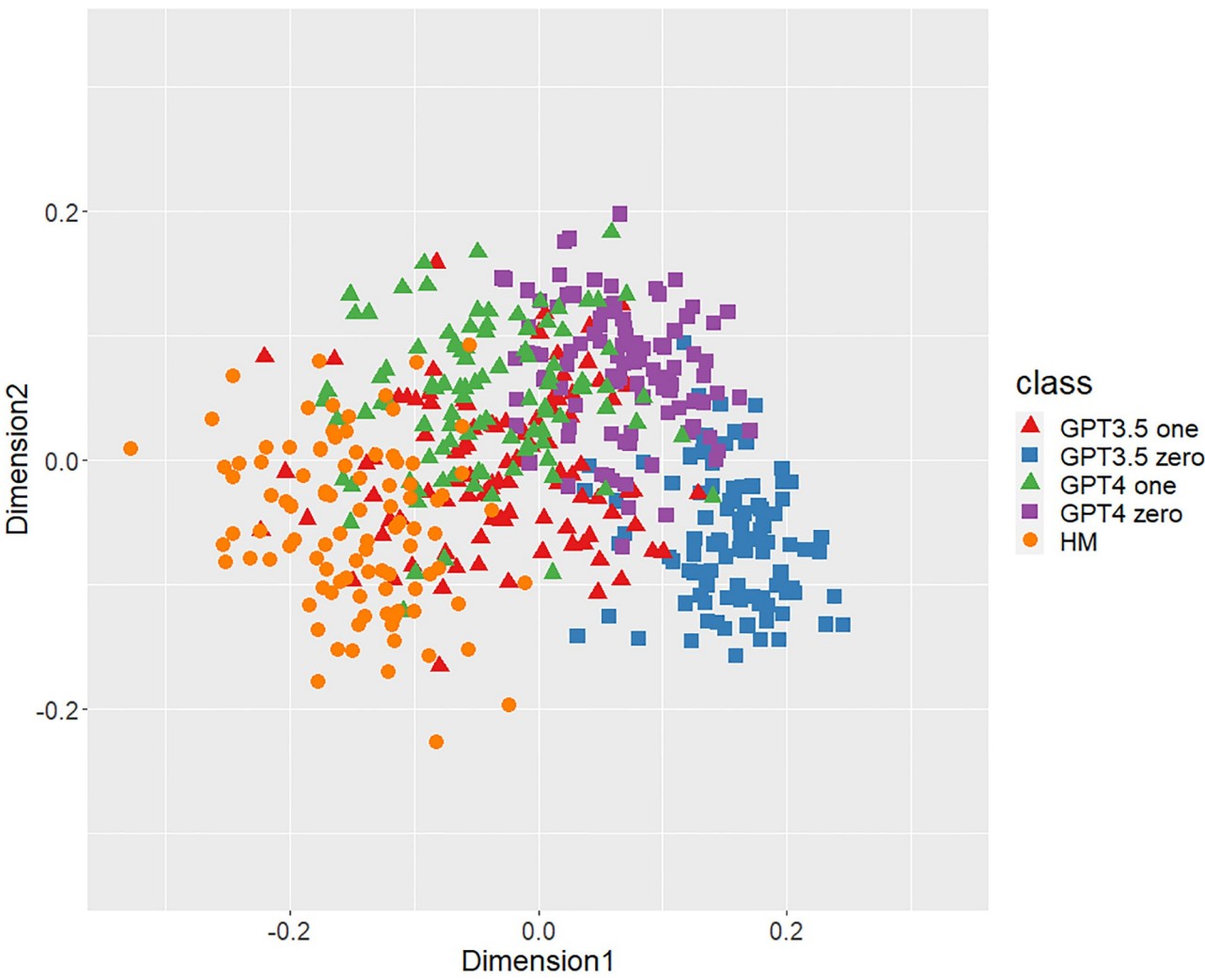

**Fig 2. MDS configuration in five classes, focusing on the POS bigrams.**

HM were shorter than those between $GPT_{zero}$ and HM. This implies that $GPT_{one}$ texts are more similar to human texts compared to $GPT_{zero}$. These results support Hypothesis 2.

Only the positioning of commas (Fig 5) displays a mixture of all classes, which means that the positioning of commas is not an effective feature for classifying ChatGPT-generated and human-written public comments. Based on the above results, we judged phrase patterns (Fig 1), POS bigrams (Fig 2), POS trigrams (Fig 3), and function words (Fig 6) to be effective stylometric features for discriminating texts between ChatGPT and humans. Therefore, we integrated these four stylometric features and used them as "integrated features" for the next analysis. Fig 7 shows the MDS configuration of the texts, focusing on integrated features.

## Study 2: Evaluation of performance of RF classifier at LOOCV

First, we integrated GPT-3.5 and GPT-4 texts in each GPT-generated type, such as the three classes ($GPT_{zero}$, $GPT_{one}$, and HM). To evaluate the performance level for classifying the three classes using RF, we executed LOOCV and created confusion matrices for multiclass

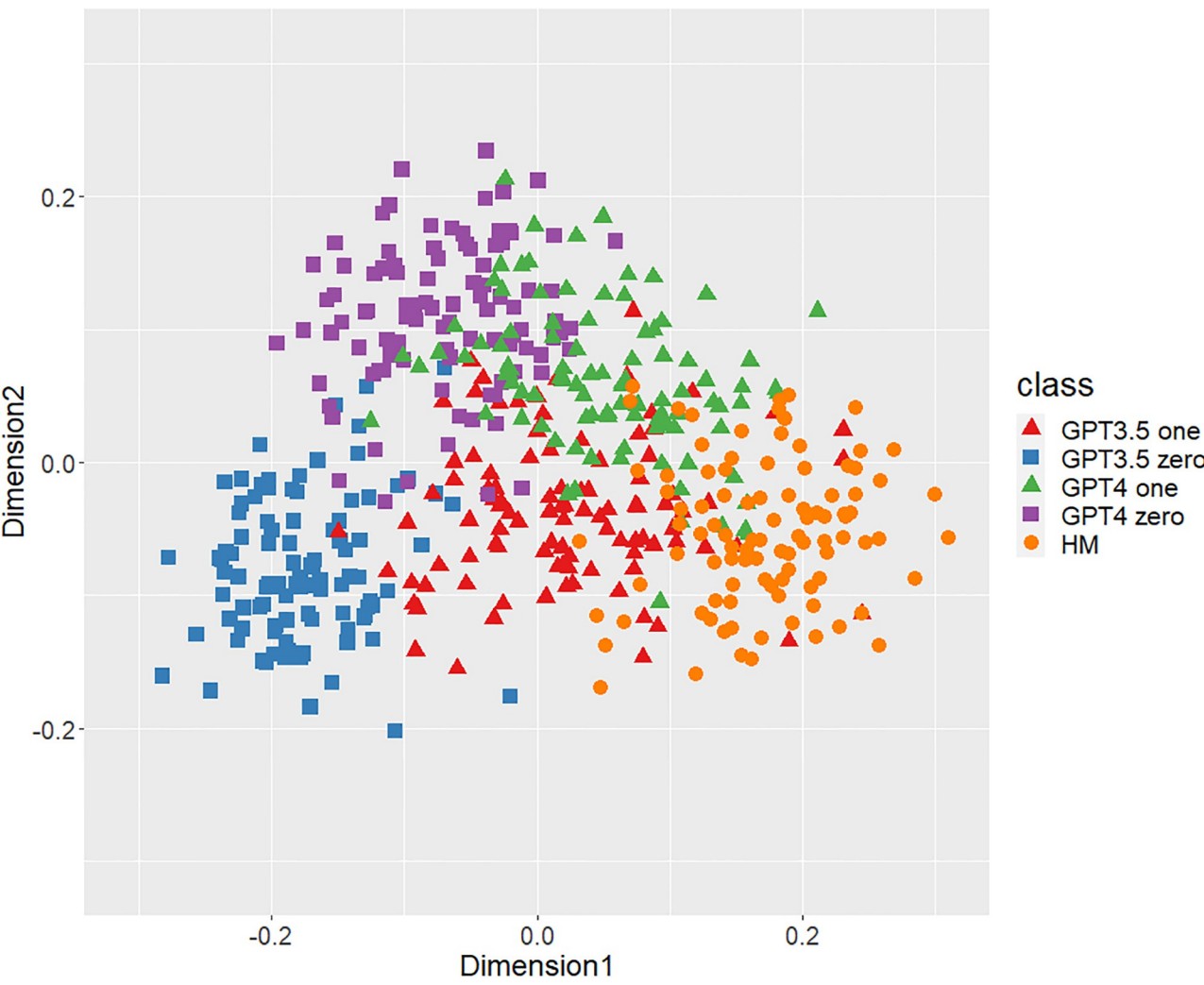

**Fig 3. MDS configuration in five classes, focusing on the POS trigrams.**

classification based on true classes and classified classes. Table 2 presents an example confusion matrix for these three classes. For instance, the cell of *a* in Table 2 means that RF classifier correctly judges text generated by ChatGPT with zero-shot learning as "GPT$_{zero}$", whereas the one of *c* indicates mistakes a judge as the text written by human. Next, based on the confusion matrix, the classification performance was assessed using the following metrics: "accuracy" in Eq (2), "recall" in Eqs (3A) to (3C), and "precision" in Eqs (4A) to (4C). The metric values were calculated for each class, together with the macro-average values (Eqs (5A) to (5B)).

$$Accuracy = \frac{a + e + i}{all\ texts\ (N = 500)} \tag{2}$$

$$Recall\ for\ GPT_{zero} = \frac{a}{a + b + c} \tag{3A}$$

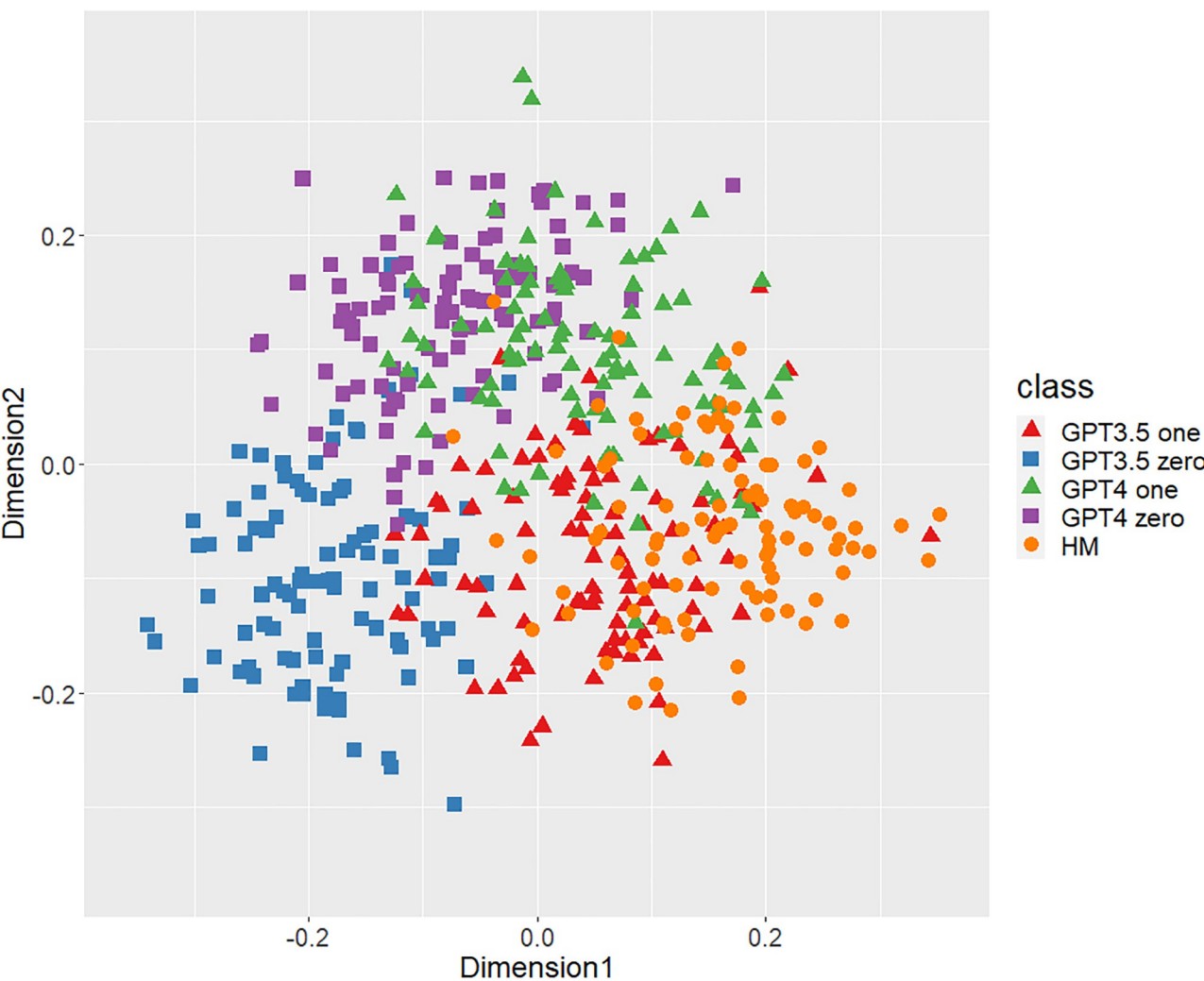

**Fig 4. MDS configuration in five classes, focusing on the bigram of postpositional particle words.**

or

$$Recall\ for\ GPT_{one} = \frac{e}{d+e+f} \qquad (3B)$$

or

$$Recall\ for\ HM = \frac{i}{g+h+i} \qquad (3C)$$

$$Precision\ for\ GPT_{zero} = \frac{a}{a+d+g} \qquad (4A)$$

or

$$Precision\ for\ GPT_{one} = \frac{e}{b+e+h} \qquad (4B)$$

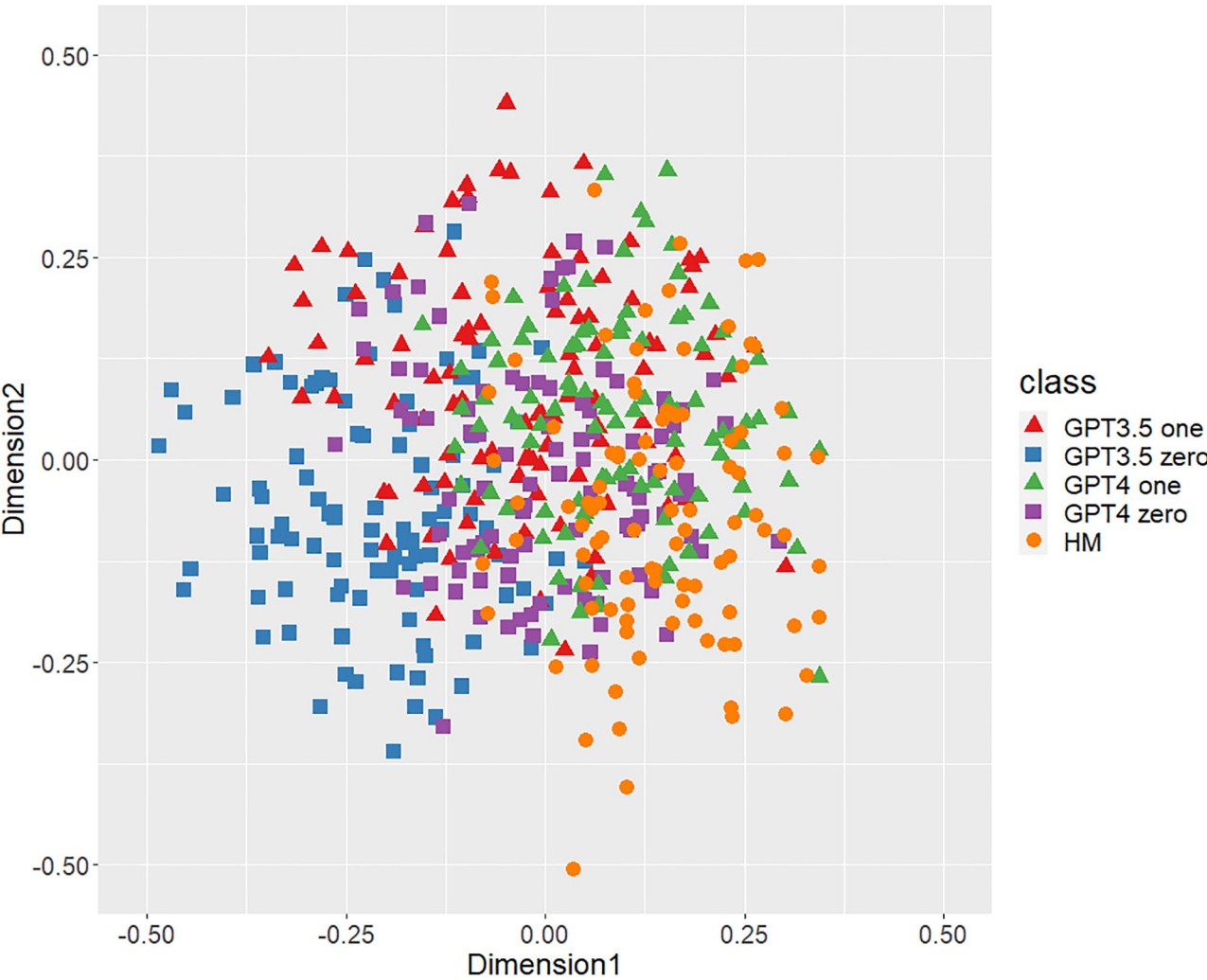

**Fig 5. MDS configuration in five classes, focusing on the positioning of commas.**

or

$$Precision\ for\ HM = \frac{i}{c + f + i} \tag{4C}$$

$$Macro\ average\ for\ recall = \frac{Recall\ for\ GPT_{zero} + GPT_{one} + HM}{3} \tag{5A}$$

$$Macro\ average\ for\ precision = \frac{Precision\ for\ GPT_{zero} + GPT_{one} + HM}{3} \tag{5B}$$

Additionally, we combined the class of $GPT_{zero}$ and $GPT_{one}$ texts as "$GPT_{zero\ and\ one}$" and calculated "recall for $GPT_{zero\ and\ one}$" and "precision for $GPT_{zero\ and\ one}$". Refer to the following Eqs (6A) and (6B) for details of the metric calculations. Among these performance metrics, we regard both "precision for HM" of Eq (4C) and "precision for $GPT_{zero\ and\ one}$" of Eq (6B) as the

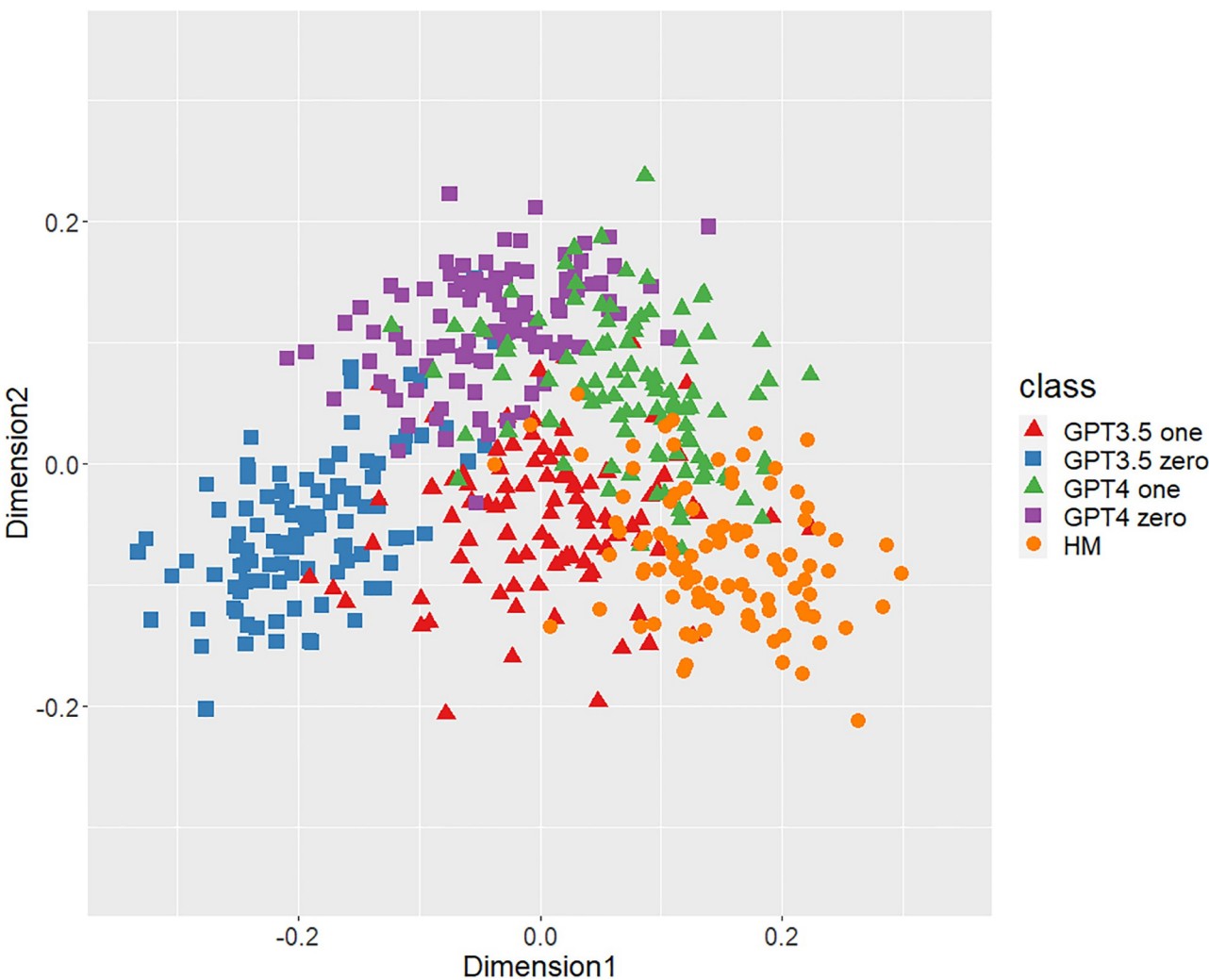

**Fig 6. MDS configuration in five classes, focusing on the function words.**

most important performance metrics because our human want to accurately predict whether the sentences by an unknown author was written by ChatGPT or by a human.

$$Recall\ for\ GPT_{zero\ and\ one} = \frac{a+b+d+e}{a+b+c+d+e+f} \tag{6A}$$

**Table 1. The means and standard deviations of distances of the entire texts between GPT (GPT$_{zero}$ and GPT$_{one}$) and HM corresponding to each stylometric feature.**

| | GPT3.5$_{zero}$ vs HM | GPT4$_{zero}$ vs HM | GPT3.5$_{one}$ vs HM | GPT4$_{one}$ vs HM |
|---|---|---|---|---|
| Phrase patterns | 0.82 (SD 0.04) | 0.79 (SD 0.04) | 0.77 (SD 0.05) | 0.75 (SD 0.05) |
| POS bigrams | 0.69 (SD 0.05) | 0.68 (SD 0.05) | 0.67 (SD 0.04) | 0.68 (SD 0.04) |
| POS trigrams | 0.94 (SD 0.03) | 0.93 (SD 0.03) | 0.92 (SD 0.04) | 0.93 (SD 0.03) |
| Bigram of postpositional particle words | 0.93 (SD 0.05) | 0.91 (SD 0.05) | 0.89 (SD 0.05) | 0.89 (SD 0.05) |
| Positioning of commas | 0.96 (SD 0.09) | 0.94 (SD 0.09) | 0.93 (SD 0.09) | 0.92 (SD 0.09) |
| Function words | 0.66 (SD 0.05) | 0.63 (SD 0.05) | 0.62 (SD 0.05) | 0.61 (SD 0.05) |

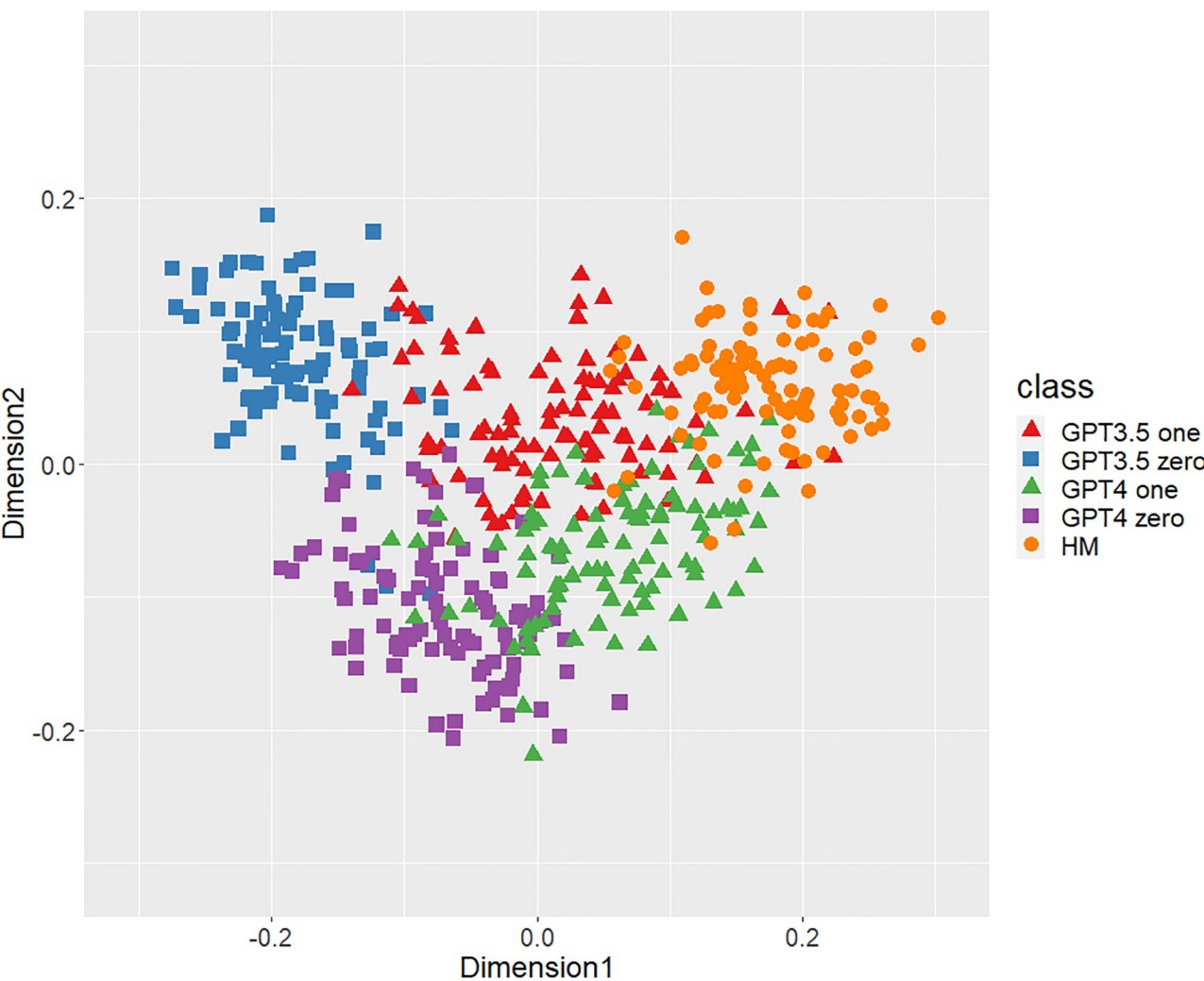

**Fig 7. MDS configuration in five classes, focusing on integrated features (the phrase patterns, the POS bigrams and trigram, and the function words).**

$$Precision\ for\ GPT_{zero\ and\ one} = \frac{a+b+d+e}{a+b+d+e+g+h} \tag{6B}$$

Table 3 is a confusion matrix for the phrase patterns and each performance metrics are follows: accuracy (90.6%), recall for $GPT_{zero}$ (95.0%), recall for $GPT_{one}$ (83.5%), recall for HM

**Table 2. Example of confusion matrix.**

| True class | Classified class | | |
|---|---|---|---|
| | $GPT_{zero}$ | $GPT_{one}$ | HM |
| $GPT_{zero}$ | a | b | c |
| $GPT_{one}$ | d | e | f |
| HM | g | h | i |

**Table 3. Confusion matrix for the phrase patterns.**

| True class | Classified class | | |
|---|---|---|---|
| | GPT$_{zero}$ | GPT$_{one}$ | HM |
| GPT$_{zero}$ | 190 | 10 | 0 |
| GPT$_{one}$ | 23 | 167 | 10 |
| HM | 1 | 3 | 96 |

(96.0%), recall for GPT$_{zero\ and\ one}$ (97.5%), precision for GPT$_{zero}$ (88.8%), precision for GPT$_{one}$ (92.8%), precision for HM (90.6%), recall for GPT$_{zero\ and\ one}$ (97.5%), precision for GPT$_{zero\ and\ one}$ (99.0%), macro average for recall (91.5%), macro average for precision (90.7%). RF classifier can suggest which variables are effective for discrimination as "importance". The importance indicated the following features are effective: "verb + れ + ます," "noun + です," and "noun + や."

Table 4 shows the confusion matrix for the POS bigrams. The results of performance metrics are as follows: Accuracy (88.0%), recall for GPT$_{zero}$ (97.5%), recall for GPT$_{one}$ (76.0%), recall for HM (93.0%), precision for GPT$_{zero}$ (87.1%), precision for GPT$_{one}$ (92.7%), precision for HM (83.0%), recall for GPT$_{zero\ and\ one}$ (95.3%), precision for GPT$_{zero\ and\ one}$ (98.2%), macro average for recall (88.8%), macro average for precision (87.6%). According to the importance of RF, "auxiliary verb +. (period)" and "postpositional particle + noun" are regarded as effective features.

The results of performance metrics calculated from confusion matrix (Table 5) for the POS trigrams: Accuracy (87.2%), recall for GPT$_{zero}$ (97.5%), recall for GPT$_{one}$ (72.0%), recall for HM (97.0%), precision for GPT$_{zero}$ (85.2%), precision for GPT$_{one}$ (94.7%), precision for HM (81.5%), recall for GPT$_{zero\ and\ one}$ (94.5%), precision for GPT$_{zero\ and\ one}$ (99.2%), macro average for recall (88.8%), macro average for precision (87.1%). According to the importance of RF, "noun + auxiliary verb +. (period)" was regarded as an effective feature. Compared with the POS bigram, the performance level decreased slightly.

Table 6 presents the confusion matrix for the bigram of postpositional particle words. The performance levels are accuracy (76.0%), recall for GPT$_{zero}$ (90.0%), recall for GPT$_{one}$ (61.0%), recall for HM (78.0%), precision for GPT$_{zero}$ (76.9%), precision for GPT$_{one}$ (75.8%), precision for HM (74.3%), recall for GPT$_{zero\ and\ one}$ (93.3%), precision for GPT$_{zero\ and\ one}$ (94.4%),

**Table 4. Confusion matrix for the POS bigrams.**

| True class | Classified class | | |
|---|---|---|---|
| | GPT$_{zero}$ | GPT$_{one}$ | HM |
| GPT$_{zero}$ | 195 | 5 | 0 |
| GPT$_{one}$ | 29 | 152 | 19 |
| HM | 0 | 7 | 93 |

**Table 5. Confusion matrix for the POS trigrams.**

| True class | Classified class | | |
|---|---|---|---|
| | GPT$_{zero}$ | GPT$_{one}$ | HM |
| GPT$_{zero}$ | 195 | 5 | 0 |
| GPT$_{one}$ | 34 | 144 | 22 |
| HM | 0 | 3 | 97 |

**Table 6. Confusion matrix for the bigram of postpositional particle words.**

| True class | Classified class | | |
|---|---|---|---|
| | GPT$_{zero}$ | GPT$_{one}$ | HM |
| GPT$_{zero}$ | 180 | 20 | 0 |
| GPT$_{one}$ | 51 | 122 | 27 |
| HM | 3 | 19 | 78 |

macro average for recall (76.3%), macro average for precision (75.7%). RF classifier indicated that "の + や", "や + の", and "や + を"are effective features.

Table 7 shows the confusion matrix for the positioning of commas. The performance levels are lower as same as bigram of postpositional particle words: Accuracy (76.6%), recall for GPT$_{zero}$ (88.0%), recall for GPT$_{one}$ (68.5%), recall for HM (70.0%), precision for GPT$_{zero}$ (78.9%), precision for GPT$_{one}$ (77.8%), precision for HM (69.3%), recall for GPT$_{zero\ and\ one}$ (92.3%), precision for GPT$_{zero\ and\ one}$ (92.5%), macro average for recall (75.5%), macro average for precision (75.4%). Importance of RF indicated "する (verb) +, (comma)" and "において (postpositional particle) +, (comma)" as effective features.

The confusion matrix for the function words is displayed in Table 8. The performance levels were relatively higher: Accuracy (88.4%), recall for GPT$_{zero}$ (95.0%), recall for GPT$_{one}$ (78.5%), recall for HM (95.0%), precision for GPT$_{zero}$ (86.0%), precision for GPT$_{one}$ (91.3%), precision for HM (88.8%), recall for GPT$_{zero\ and\ one}$ (97.0%), precision for GPT$_{zero\ and\ one}$ (98.7%), macro average for recall (89.5%), macro average for precision (88.7%). The importance of RF indicated "や (postpositional particle)" and "です (auxiliary verb)" as effective features.

Finally, we integrated four effective features (the phrase patterns, the POS bigrams and trigrams, and the function words) and analyzed them using the integrated features. Table 9 presents the confusion matrix for the integrated features. The performances were slightly improved, compared to other features: Accuracy (91.6%), recall for GPT$_{zero}$ (97.0%), recall for GPT$_{one}$ (83.0%), recall for HM (98.0%), precision for GPT$_{zero}$ (89.8%), precision for GPT$_{one}$ (95.4%), precision for HM (89.1%), recall for GPT$_{zero\ and\ one}$ (97.0%), precision for GPT$_{zero\ and\ one}$ (99.5%), macro average for recall (92.7%), and macro average for precision (91.4%). This

**Table 7. Confusion matrix for the positioning of commas.**

| True class | Classified class | | |
|---|---|---|---|
| | GPT$_{zero}$ | GPT$_{one}$ | HM |
| GPT$_{zero}$ | 176 | 18 | 6 |
| GPT$_{one}$ | 38 | 137 | 25 |
| HM | 9 | 21 | 70 |

**Table 8. Confusion matrix for the function words.**

| True class | Classified class | | |
|---|---|---|---|
| | GPT$_{zero}$ | GPT$_{one}$ | HM |
| GPT$_{zero}$ | 190 | 10 | 0 |
| GPT$_{one}$ | 31 | 157 | 12 |
| HM | 0 | 5 | 95 |

**Table 9. Confusion matrixes for the integrated features (GPT$_{zero}$ vs GPT$_{one}$ vs HM).**

| True class | Classified class | | |
|---|---|---|---|
| | GPT$_{zero}$ | GPT$_{one}$ | HM |
| GPT$_{zero}$ | 194 | 6 | 0 |
| GPT$_{one}$ | 22 | 166 | 12 |
| HM | 0 | 2 | 98 |

study demonstrated incremental validity because the integrated features achieved the best classification performance.

For reference, the mean accuracies by 10-fold cross-validation showed next: (1) the phrase patterns: 86.0% (*SD* 3.9%), (2) the POS bigrams: 84.4% (*SD* 4.0%), (3) the POS trigrams: 82.4% (*SD* 4.7%), (4) the bigram of postpositional particle words: 75.4% (*SD* 6.7%), (5) the positioning of commas: 73.4% (*SD* 4.4%), (6) the function words: 83.2% (*SD* 3.8%), (7) the integrated features: 88.0% (*SD* 3.0%).

In addition to above the analyses, we calculated the classification performance metrics by focusing only on the integrated features to compare each GPT type to HM as follows: (1) GPT$_{zero}$ (GPT 3.5$_{zero}$ vs. GPT4$_{zero}$) vs. HM and (2) GPT$_{one}$ (GPT3.5$_{one}$ vs. GPT4$_{one}$) vs. HM. With regard to GPT$_{zero}$ vs. HM, we can completely distinguish the GPT$_{zero}$ texts from the HM (Table 10). Therefore, all performance metrics (accuracy, recall, and precision for GPT$_{zero}$ vs. humans) resulted in 100%. However, in the case of GPT$_{one}$ vs. human (Table 11), the classification performance slightly decreased compared to the other cases (GPT$_{zero}$ vs. human) but maintained a high performance level: accuracy (95.3%), recall for GPT$_{one}$ (94.5%), recall for HM (97.0%), precision for GPT$_{one}$ (98.4%), and precision for HM (89.8%).

## Discussion

This study examined whether we could distinguish between human public comments and ChatGPT-generated fake public comments (including ChatGPT-emulated humans) using Japanese stylometric analysis.

According to Study 1, the results of the MDS indicated that GPT$_{zero}$ texts generated by presenting only the titles of public comments applicable to zero-shot learning were completely

**Table 10. Confusion matrixes for the integrated features (GPT 3.5$_{zero}$ vs. GPT4$_{zero}$ vs. HM).**

| True class | Classified class | | |
|---|---|---|---|
| | GPT 3.5$_{zero}$ | GPT4$_{zero}$ | HM |
| GPT 3.5$_{zero}$ | 98 | 2 | 0 |
| GPT4$_{zero}$ | 3 | 97 | 0 |
| HM | 0 | 0 | 100 |

**Table 11. Confusion matrixes for the integrated features (GPT3.5$_{one}$ vs GPT4$_{one}$ vs HM).**

| True class | Classified class | | |
|---|---|---|---|
| | GPT3.5$_{one}$ | GPT4$_{one}$ | HM |
| GPT3.5$_{one}$ | 92 | 2 | 6 |
| GPT4$_{one}$ | 5 | 90 | 5 |
| HM | 3 | 0 | 97 |

different from human-written texts. However, most of the $GPT_{one}$ texts, which emulated human public comments (i.e., one-shot learning), were positioned between the distributions of $GPT_{zero}$ and HM on the MDS dimension. Furthermore, some $GPT_{one}$ texts overlapped slightly with the human texts. These results support Hypotheses 1 and 2: Japanese stylometric features of $GPT_{zero}$ texts are completely different from those of human public comments, and $GPT_{one}$ texts are more similar to human public comments than $GPT_{zero}$. We consider that this center positioning of the $GPT_{one}$ texts means not "closer from $GPT_{zero}$ to human" but "closer from human to $GPT_{zero}$" because $GPT_{one}$ may start emulating and generating from human public comment. That is, $GPT_{one}$ texts may be closer to $GPT_{zero}$ texts by emulating and modifying the HM texts. Furthermore, according to the Figure (especially Figs 1–3), the texts of $GPT4_{one}$ are farther away from the distribution of HM texts than $GPT3.5_{one}$. These results suggest that the higher the performance of ChatGPT (i.e., GPT-4 at present), the easier it may be to distinguish emulated texts from human-written texts because higher-performance ChatGPT can more sophisticatedly rewrite human-written texts to make them closer to $GPT_{zero}$ texts. Regardless of the lower word counts in the current study (appropriately 600 characters vs. 1,000 characters in a previous study [5]), the differences between the GPT with zero-shot learning and humans were larger in the current study than in the previous study. It is unclear why these results occurred because several factors, such as word count (600 words vs. 1,000 words) and categories (public comments vs. academic papers), were confounded. Fig 5 indicates that the positioning of commas had little distinguishable effect because almost all texts in each class overlapped. A previous study [5] demonstrated a certain effective level of comma positioning. We considered the possibility that the difference in genres (academic papers and public comments) influenced these results. Therefore, we need to further examine other genres of texts.

Study 2 showed that the best precision HM achieved was approximately 90% and that $GPT_{zero\ and\ one}$ reached was 99.5%. Considering these results, it can be said that Hypothesis 3 was supported: we can discriminate fake public comments generated by ChatGPT from human public comments. Among the six Japanese stylometric features, phrase patterns indicated the best discriminable performance and POS bigrams and trigrams showed high classification accuracy. ChatGPT is not good at rewriting texts taking these features into consideration because these stylometric features (the phrase patterns, POS bigrams, and POS trigrams) are regarded as a deeper structural aspect of sentences. However, the present study revealed low performance of the positioning of commas, particularly in the GPT emulation. ChatGPT can easily rewrite this feature in sentences because of linguistically low-level features. While presenting human public comments and making ChatGPT emulate, we confirmed ChatGPT often just paraphrased words (e.g., from "ignorant" to "fool"). Therefore, presently, even if we analyze other languages, we may be able to distinguish sentences between generative AI and humans by focusing on deeper structures.

Above the results of this study limited Japanese language. Zaitsu & Jin [5] also pointed out that Japanese language have different notation formats (Kanji, Hiragana, and Katakana) and no space between words as opposed to English. Therefore, we need conduct similar verification for other languages as well. In addition, we need collect and analyze larger sample size of human-written and AI-generated public comments for the purpose of generalization of this study.

Recently, the disinformation generated by AI, such as fake news, has become a problem worldwide because these fakes are instantly and widely generated. Disinformation has certainly caused chaos in the human world; therefore, we need techniques to control generative AI, including sophisticated classifiers.

## Conclusion

The current study concluded that (1) the stylometric features of Japanese public comments were completely different from ChatGPT-generated texts by presenting only the titles of public comments (i.e., zero-shot learning). (2) The public comments generated by the one-shot trained ChatGPT with human-generated public comments are more similar to human public comments than the public comments from the zero-shot trained ChatGPT. (3) Although limited to this study sample (Japanese language, approximately 600 characters, and learning method of ChatGPT), at present, we can discriminate ChatGPT-generated fake public comments from human public comments through stylometric analysis.

## Supporting information

**S1 Data.**
(CSV)

**S2 Data.**
(CSV)

**S3 Data.**
(CSV)

**S4 Data.**
(CSV)

**S5 Data.**
(CSV)

**S6 Data.**
(CSV)

## Author Contributions

**Conceptualization:** Wataru Zaitsu, Mingzhe Jin, Shunichi Ishihara, Satoru Tsuge.

**Formal analysis:** Wataru Zaitsu, Mingzhe Jin.

**Funding acquisition:** Wataru Zaitsu, Mitsuyuki Inaba.

**Investigation:** Wataru Zaitsu.

**Methodology:** Wataru Zaitsu, Mingzhe Jin, Shunichi Ishihara, Satoru Tsuge.

**Project administration:** Mingzhe Jin.

**Resources:** Wataru Zaitsu, Shunichi Ishihara.

**Software:** Mingzhe Jin.

**Supervision:** Mingzhe Jin, Shunichi Ishihara.

**Visualization:** Wataru Zaitsu.

**Writing – original draft:** Wataru Zaitsu.

**Writing – review & editing:** Wataru Zaitsu, Shunichi Ishihara, Satoru Tsuge, Mitsuyuki Inaba.

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
