## [Decision Letter · Decision Letter 0]

11 Dec 2023

PONE-D-23-32297Can we spot fake public comments generated by ChatGPT(-3.5, -4)?: Japanese stylometric analysis expose emulation through one-shot learningPLOS ONE

Dear Dr. Zaitsu,

Thank you for submitting your manuscript to PLOS ONE. After careful consideration, we feel that it has merit but does not fully meet PLOS ONE’s publication criteria as it currently stands. Therefore, we invite you to submit a revised version of the manuscript that addresses the points raised during the review process.

We look forward to receiving your revised manuscript.

Kind regards,

Takayuki Mizuno, Ph. D.

Academic Editor

PLOS ONE

Journal Requirements:

"This work was partially supported by JSPS KAKENHI (grant number: JP23K11107). The funders had no role in the study design, data collection, analysis, and decision to publish, except for the Publication Fee."

"This work was partially supported by JSPS KAKENHI (grant number: JP23K11107). The funders had no role in the study design, data collection, analysis, and decision to publish, except for the Publication Fee."

5. Please upload a copy of Figure 1, to which you refer in your text on page 9. If the figure is no longer to be included as part of the submission please remove all reference to it within the text.

Additional Editor Comments:

Please revise your manuscript according to the reviewers' comments.

Reviewers' comments:

Reviewer's Responses to Questions

**Comments to the Author**

1. Is the manuscript technically sound, and do the data support the conclusions?

Reviewer #1: Yes

Reviewer #2: Partly

2. Has the statistical analysis been performed appropriately and rigorously? 

Reviewer #1: Yes

Reviewer #2: Yes

3. Have the authors made all data underlying the findings in their manuscript fully available?

Reviewer #1: Yes

Reviewer #2: Yes

4. Is the manuscript presented in an intelligible fashion and written in standard English?

Reviewer #1: Yes

Reviewer #2: Yes

5. Review Comments to the Author

Reviewer #1: Overall, this is exciting work, but significant improvement is required in this manuscript. Some of the improvement requirements are as follows:

1. Motivation and Contribution of the work need to be incorporated in the sub-section of the Introduction. Further, at the end of the Introduction section, the structure of the rest of the manuscript work should be incorporated.

2. All the equations should be written along with the equation number and need to be appropriately cited.

3. Further each variable used in the equation needs to be discussed properly.

4. In this work, in the study 2 2-section Random Forest (ML) algorithm was issued. Here, preprocessing steps need to be discussed further 499 data is taken for the experiment. What is the training and testing ratio?

5. What is the reason for selecting Random Forest Algorithm? There are several other algorithms.

6. What is the limitation of this work? Limitations of this work need to be incorporated.

7. Whether in this work “emoji” and “sarcasm” are considered?

8. Proper capstion is required for the figures used in this manuscript.

Reviewer #2: The topic is interesting and tackles an important issue. However, there are some concerns which can be summarized as follows.

1- The sample size of the data is too small to be able to conclude definitive conclusions.

2- Full details of the conferences cited in the paper should be given. This means that the place where the conference was held should be given.

6. PLOS authors have the option to publish the peer review history of their article (what does this mean?). If published, this will include your full peer review and any attached files.

Reviewer #1: No

Reviewer #2: **Yes: **Ahmed Sharaf Eldin

---

## [Author Response · Author response to Decision Letter 0]

19 Dec 2023

Answer and Response to Editor and Reviewer 1 comments:

Thank you for your constructive review.

I revised several points along with your comments.

Please confirm revised paper. 

Thank you.

1. Additional explain of contribution of this study in the Introduction. 

 We additionally explained contribution of this study (p4, l80).

2. equations number and citation

 We give number in all equations and appropriately cited.

3. “Further each variable used in the equation needs to be discussed properly.”

 Sorry. We don’t understand this comment. Would you explain this more.

4. The training and testing ratio

 We conducted LOOCV. Among 500 sample, we used 499 sample for training and 1 sample for test. In this study, we did not set validation sample because we fixed hyperparameters. 

5. About Random Forest Algorithm

 We politely explained why we used random forest (p8, l192). 

6. The limitation of this study.

 We added the limitation of this study in discussion, along with reviewer’s comment (p20, l406).

7. emoji and sarcasm

 There was no emoji in our public comments. Usually, “emoji” are used in informal circumstances, for example, exchange between acquaintances. On the other hand, public comments are formal circumstances.

 Furthermore, we did not confirm“sarcasm”. Moreover, we focused on non-content words, so this stylometric analysis is not affected by sarcasm. 

　8. Capstion in the figures

We revised and added caption explain.

Answer and Response to Reviewer 2 comments:

Thank you for your constructive review.

I revised several points along with your comments.

Please confirm revised paper. 

Thank you.

1.The limitation of this study.

 We added the limitation of this study in discussion, along with reviewer’s comment (p20, l406).

2.Revised references

 We revised references, especially added details about conferences, along with reviewer’s comment.

---

## [Decision Letter · Decision Letter 1]

30 Jan 2024

PONE-D-23-32297R1Can we spot fake public comments generated by ChatGPT(-3.5, -4)?: Japanese stylometric analysis expose emulation through one-shot learningPLOS ONE

Dear Dr. Zaitsu,

Thank you for submitting your manuscript to PLOS ONE. After careful consideration, we feel that it has merit but does not fully meet PLOS ONE’s publication criteria as it currently stands. Therefore, we invite you to submit a revised version of the manuscript that addresses the points raised during the review process.

We look forward to receiving your revised manuscript.

Kind regards,

Takayuki Mizuno, Ph. D.

Academic Editor

PLOS ONE

Journal Requirements:

Additional Editor Comments:

Reviewer 1 commented that the manuscript still needs some revision. Please add comments on the ratio of training data to test data and the variables noted by reviewer 1.

Reviewers' comments:

Reviewer's Responses to Questions

**Comments to the Author**

1. If the authors have adequately addressed your comments raised in a previous round of review and you feel that this manuscript is now acceptable for publication, you may indicate that here to bypass the “Comments to the Author” section, enter your conflict of interest statement in the “Confidential to Editor” section, and submit your "Accept" recommendation.

Reviewer #1: All comments have been addressed

Reviewer #2: All comments have been addressed

2. Is the manuscript technically sound, and do the data support the conclusions?

Reviewer #1: Partly

Reviewer #2: Yes

3. Has the statistical analysis been performed appropriately and rigorously? 

Reviewer #1: No

Reviewer #2: Yes

4. Have the authors made all data underlying the findings in their manuscript fully available?

Reviewer #1: No

Reviewer #2: No

5. Is the manuscript presented in an intelligible fashion and written in standard English?

Reviewer #1: Yes

Reviewer #2: Yes

6. Review Comments to the Author

Reviewer #1: Some of the suggested improvements are positively addressed by the authors. But there are some more points to be positvely addressed.

1. Out of 500 sample , 499 is taken for training and 1 is taken for testing. Standard traing and testing ratio is 80-20, 70-30. In the current mauscript case the model will go in the overfitting condtion, so k-fold cross validation is required to check the overfitting condtion.

2. Each variable/parameters used in equation need to be disccused. Example in equation 6A, (parameters a,b,c,d,e,f) need to be discussd. What is a, what is b , like wise.

Reviewer #2: The authors reviewed their paper according to the reviewers' comments. It will be useful if the authors discuss the computational resources required to do their experiments.

7. PLOS authors have the option to publish the peer review history of their article (what does this mean?). If published, this will include your full peer review and any attached files.

Reviewer #1: **Yes: **sanjay kumar

Reviewer #2: **Yes: **Ahmed Sharaf Eldin

---

## [Author Response · Author response to Decision Letter 1]

2 Feb 2024

Answer and Response to Editor and Reviewer 1 comments:

Thank you for your constructive review.

I revised several points along with your comments.

Please confirm revised paper. 

Thank you.

1. Out of 500 sample , 499 is taken for training and 1 is taken for testing. Standard traing and testing ratio is 80-20, 70-30. In the current mauscript case the model will go in the overfitting condtion, so k-fold cross validation is required to check the overfitting condtion.

LOOCV (Leave-one-out cross-validation), conducted in this study, is generally less prone to overfitting compared to other cross-validation methods like k-fold cross-validation, especially when the dataset is small like this current study. Therefore, we used this method.

But for reference, we conducted 10-fold cross validation, and wrote these results.

2. Each variable/parameters used in equation need to be disccused. Example in equation 6A, (parameters a,b,c,d,e,f) need to be discussd. What is a, what is b , like wise.

Along with this comments, we additionally explain in details (p16, l377). 

3.Change part of the title

Before this submission, the title is ““Can we spot fake public comments generated by ChatGPT(-3.5, -4)?: Japanese stylometric analysis expose emulation through one-shot learning”. But I think “through one-shot” may lead to misunderstanding for readers that not “expose emulation created by one-shot learning” but “expose by one-shot learning”. So we want to change part of the title to “Can we spot fake public comments generated by ChatGPT(-3.5, -4)?: Japanese stylometric analysis expose emulation created by one-shot learning”

---

## [Editor Report · Decision Letter 2]

5 Feb 2024

Can we spot fake public comments generated by ChatGPT(-3.5, -4)?: Japanese stylometric analysis expose emulation created by one-shot learning

PONE-D-23-32297R2

Dear Dr. Zaitsu,

We’re pleased to inform you that your manuscript has been judged scientifically suitable for publication and will be formally accepted for publication once it meets all outstanding technical requirements.

Kind regards,

Takayuki Mizuno, Ph. D.

Academic Editor

PLOS ONE

Additional Editor Comments (optional):

The authors have incorporated all the reviewers' comments into the revised manuscript.
---

## [Editor Report · Acceptance letter]

20 Feb 2024

PONE-D-23-32297R2 

PLOS ONE

Dear Dr. Zaitsu, 

I'm pleased to inform you that your manuscript has been deemed suitable for publication in PLOS ONE. Congratulations! Your manuscript is now being handed over to our production team.

Kind regards, 

on behalf of

Dr. Takayuki Mizuno 

Academic Editor

PLOS ONE